# The Historical Development of Infrared Photodetection Based on Intraband Transitions

**DOI:** 10.3390/ma16041562

**Published:** 2023-02-13

**Authors:** Qun Hao, Xue Zhao, Xin Tang, Menglu Chen

**Affiliations:** 1School of Optics and Photonics, Beijing Institute of Technology, Beijing 100081, China; 2Beijing Key Laboratory for Precision Optoelectronic Measurement Instrument and Technology, Beijing 100081, China; 3Yangtze Delta Region Academy of Beijing Institute of Technology, Jiaxing 314019, China

**Keywords:** intraband transition, infrared photodetection, quantum well, quantum dots

## Abstract

The infrared technology is entering widespread use as it starts fulfilling a growing number of emerging applications, such as smart buildings and automotive sectors. Majority of infrared photodetectors are based on interband transition, which is the energy gap between the valence band and the conduction band. As a result, infrared materials are mainly limited to semi-metal or ternary alloys with narrow-bandgap bulk semiconductors, whose fabrication is complex and expensive. Different from interband transition, intraband transition utilizing the energy gap inside the band allows for a wider choice of materials. In this paper, we mainly discuss the recent developments on intraband infrared photodetectors, including ‘bottom to up’ devices such as quantum well devices based on the molecular beam epitaxial approach, as well as ‘up to bottom’ devices such as colloidal quantum dot devices based on the chemical synthesis.

## 1. Introduction

In recent years, infrared detectors are widely used in applications such as imaging, medical, and security [1,2,3], where achieving multi-color, large-scale array, high performance, and low cost are the primary goals. Majority of infrared optoelectronic devices are composed of semiconductors utilizing interband transition (transition of electrons from the valence band to the conduction band) [4]. Due to the low photon energy of infrared light, the bandgap of corresponding infrared materials is required to be narrow. Consequently, infrared materials based on interband transitions are typically restricted to narrow-band semiconductor materials, such as group III–V, group IV–VI, and group II–VI [5,6,7] materials, which are mainly based on molecular epitaxy and need to be grown on suitable substrates under a high-vacuum environment. These processes are complex, challenging, have low yield, and are expensive.

In contrast, the photodetector via intraband transition (transition of electronic states inside the band) could provide a solution to break through the bandgap limitation [8]. The intraband transition would make it possible to realize the infrared photodetection of wide-band semiconductors, which expands the infrared materials’ growth techniques. Furthermore, the intraband infrared photoluminescence decays show absent or greatly reduced Auger relaxation compared with the interband, due to the sparse density of states in the conduction band [9]. This feature would benefit the quantum efficiency in infrared photodetectors.

The major intraband transition-based photodetectors include ‘up to bottom’ devices, such as quantum well infrared photodetectors (QWIPs), and the emerging ‘bottom to up’ devices such as colloidal quantum dot (CQD) photodetectors. The infrared materials in QWIPs are grown based on the molecular beam epitaxy method. The technology of such devices is quite mature and has already been commercialized [10,11]. The dark current is low in QWIP, benefitting from the homogeneity of the quantum well. Besides, compared with HgCdTe bulk materials [12], it is simpler to fabricate larger-scale wafers on the quantum well. Still, there are disadvantages in QWIPs, such as low quantum efficiency, low operating temperature, the cumbersome production process, and that they cannot directly probe vertically incident radiation due to the transition selection rule [13]. Compared with the quantum well, the thermal carrier generation would be significantly reduced in quantum dots (QDs), which may benefit the high operating temperature. More importantly, there has been great progress in the QDs’ growth methods in the last few decades. For example, CQDs are distinguished by an obvious quantum confinement effect, tunable size and bandgap, excellent photostability, and high integration [14,15,16,17,18,19]. The CQDs based on intraband transitions of absorption or emission can cover the entire infrared band and have spectral tunability [20,21,22,23], which have great advantages compared with other photoelectric materials. In Figure 1, we mainly present QWIPs and quantum dots infrared photodetectors (QDIPs) based on intraband transition in recent years.

In this paper, we will discuss the materials and device properties of intraband quantum well and quantum dot devices. Firstly, we introduce epitaxial growth InAs- and SiGe-based infrared intraband photodetectors. Then, we discuss the ‘bottom to up’ intraband photodetectors based on the emerging semiconductor CQDs. We also report the development status of perovskite-based intraband transition devices. Finally, we summarize the intraband transition infrared detectors, and look to the future of these devices.

## 2. Infrared Photodetectors Based on Intraband Transition

### 2.1. ‘Up to Bottom’ Infrared Intraband Photodetector

#### 2.1.1. InAs-Based Infrared Intraband Photodetector

QWIPs based on intraband transitions in quantum heterostructures are widely used in the infrared subregions, such as the mid-infrared band (3–5 μm), long infrared band (8–12 μm), and far-infrared band (>12 μm). The infrared semiconductor devices of quantum wells are mainly based on InAs.

In 1993, B.F. Levine researched the intraband absorption and transport processes based on InGaAs QWIPs [24]. To widen the range of the probe, Fabio Durante Pereira Alves et al. studied three-band AlGaAs/GaAs/InGaAs QWIPs based on both interband and intraband transition (Figure 2a) [25]. Three peaks were given at wavelengths of 0.84, 5.0, and 8.5 µm, and the responsivities could reach 0.5 A/W, 0.03 A/W, and 0.13 A/W, respectively. This result made it possible to widen the detection range of quantum well devices.

In 2011, Unil Perera et al. designed InGaAs/GaAs/AlGaAs npn-QWIPs (Figure 2b), with dual-band absorption [26]. They measured the intraband response, where the intraband response for positive bias (two peaks at 3.6 and 6.0 µm) was from the InGaAs well, and the response for negative bias (peak at 10.8 µm) was from GaAs wells. The specific detectivity values reached 2 × 10^8^ Jones and 5 × 10^9^ Jones for 3.6 and 10.8 μm at the temperature of 80 K. At the same time, they designed the structure of the quantum ring intraband detector as shown in Figure 2c. The responsivity was 20 or 25 A/W at 165 μm (1.8 THz) and the detectivity values were 1 × 10^16^ Jones at 5.2 K and 3 × 10^15^ Jones at 10 K. The response time of the quantum ring intraband detectors was 300 ps. The development of dual-band QWIPs opens a new direction for the development of quantum well devices.

To realize the application of large-array detection, Yetkin Arslan et al. suggested a large-format focal plane array (640 × 512) based on InP/In_0.48_Ga_0.52_As QWIPs in 2013 [27].

Since the InAs/GaAs QWIP had the “g-r” (generation-recombination) noise caused by the combination of electrons and holes, Xiangfei Wei et al. optimized the InAs/GaSb quantum well structure by adding an AlSb cap layer in 2020 [28], which effectively reduced the “g-r” noise generated by the combination of electrons and holes (Figure 2d). When the width of the AlSb film reached 2 nm, the optical transitions of carriers in different material layers were significantly reduced, and thus the “g-r” noise could be reduced, and the detector worked well at room temperature. In the same year, Heming Yang et al. presented the effect of temperature in molecular beam epitaxy growth methods [29]. The responsivity of In_0.14_Ga_0.86_As/GaAs QWIPs grown at a low temperature increased 38 times, reaching 5.67 A/W at 20 K, compared with In_0.14_Ga_0.86_As/GaAs QWIPs created by a temperature-changed growth method (Figure 2e,f).

The narrow-band performance of pixel-level photodetectors was used in gas detection, and the performance of the photodetector faces great challenges. Therefore, several researchers have improved the performance of intraband devices by adding an optical structure. In 2022, Tianyang Dong et al. created pixel-level, narrow-band, high quantum efficiency QWIPs with metal microcavity at 10.6 μm [30]. The device structure of Al_0.5_Ga_0.5_As/n^+^ GaAs is shown in Figure 2g. The optical coupling efficiency peaked when the cavity height was 3.1 μm. By changing the width of the microcavity, several different resonance modes could be obtained, whose peak wavelength closed to the intrinsic detection wavelength. As depicted in Figure 2i, the quantum absorption efficiency of the device was greater than 60% in three different resonant modes. In addition, the device responsivity was 1.2 A/W at 10.35 μm. Notably, the peak responsivity of the 45° standard reference device was only 0.127 A/W at 10.33 μm (Figure 2j). The structure of the Al_0.5_Ga_0.5_As/n^+^ GaAs QWIP was suitable for narrow-band gas detection and multi-spectral detection due to its high peak responsivity and narrow bandwidth.

**Figure 2 materials-16-01562-f002:**
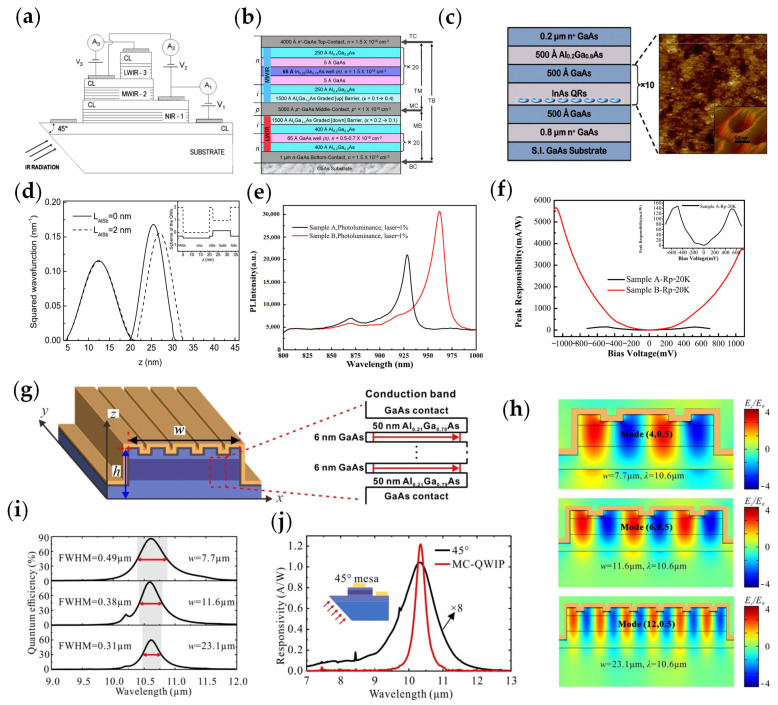
Structure and performance characterization of InAs−based infrared intraband photodetectors: (**a**) Schematic of AlGaAs/GaAs/InGaAs QWIPs [25]. Copyright 2008, Journal of Applied Physics. (**b**) Structure diagram of the n−p−n InGaAs/GaAs/AlGaAs QWIP. The QWIP consisted of three doped layers (top contact, middle contact, and bottom contact) and an active region (mid−wave−infrared and long−wave−infrared). (**c**) Structure diagram of quantum ring intraband detector [26]. Copyright 2011, 11th IEEE International Conference on Nanotechnology. (**d**) Ground−state wave functions for electrons and holes at different widths of AlSb. The solid line represents functions at widths of 0 nm and the dashed line represents functions at widths of 2 nm [28]. Copyright 2020, Physica E: Low−dimensional Systems and Nanostructures. (**e**) Photoluminescence spectra for InGaAs/GaAs QWIPs with different growth methods, including the temperature−changed growth method (sample A) and the low−temperature growth method (sample B). (**f**) Responsivities of the two samples under different bias voltages. The inset was the peak responsivity of sample A [29]. Copyright 2020, Journal of physics. D, Applied physics. (**g**) Schematic diagram of the Al_0.5_Ga_0.5_As/n^+^ GaAs QWIP and the band structure of the QWIP material. (**h**) Simulation of the E_z_/E_0_ (E_0_: x−polarized plane electromagnetic wave, E_z_: z−polarized plane electromagnetic wave) field distribution at the resonance wavelength for different mesa widths of the QWIP. (**i**) Simulated quantum efficiency with different mesa. (**j**) Spectral responsivity of QWIP and 45° standard device at 40 K with 2 V bias [30]. Copyright 2022, Applied Physics Letters.

From the above analysis, it is obvious that quantum well devices with intraband transition have developed rapidly in the last several decades, and Table 1 summarizes the development. Focal plane array based on intraband QWIPs has also been investigated, which should be the future trend [27].

#### 2.1.2. SiGe-Based Infrared Intraband Photodetector

In addition to the InAs-based intraband transition devices, SiGe-based intraband transition devices have also been developed.

In 2001, C. Miesner et al. proposed intraband transition lateral photodetectors based on self-assembled Ge dots by molecular beam epitaxy, which could work in 3.3 and 4.4 μm [41]. The responsivity reached a maximum of 10 mA/W (Figure 3a) and the detectivity reached 1 × 10^11^ Jones at the temperature of 20 K. In 2003, Dominique Bougeard et al. investigated the intra-valence band photoexcitation of holes from self-assembled Ge QDs in Si, which reached a maximum response in the mid-wavelength range of 3–5 μm [42]. The introduction of the SiGe channel within the active structure region increased the responsivity. The photoresponsivity of this structure can be achieved at 90 mA/W for a 3.6 μm wavelength at the temperature of 20 K. In Figure 3b, the photocurrent spectra of a Si/Ge/Si/SiGe multilayer structure were studied. The maximum photocurrent in the quantum well structure was attributed to transitions from Ge dot ground states to quasi-bound states at the Si valence band edge. The result showed that Si/Ge quantum well structures with lateral optical detection would be expected to be a high-sensitivity and large-area mid-infrared photodetector.

To further research the relationship between temperature and the photocurrent response, R. K. Singha et al. studied the room temperature photocurrent response of Ge/Si self-assembled QDs in the wavelength range of 3.1 μm and 6.2 μm [43]. The results showed that the photocurrent was red-shifted with the increasing temperature (Figure 3c,d).

In 2011, Patrick Rauter et al. reported the fabrication and characterization of tensile-strained p-type SiGe QWIPs grown on a Si_0.74_Ge_0.26_ pseudo-substrate [44]. The QWIPs worked from a light-hole (LH) ground state and had a response in both the mid-infrared and terahertz regions. Experiments showed that the device had good performance, which was 5.7 × 10^7^ Jones at a bias of 0.5 V. The responsivity peaks of values in terahertz and mid-infrared regimes (Figure 3f) were approximately 30 meV (LH-LH, 0.3 mA/W at 0.5 V), 85 meV (LH-HH (heavy hole), 0.4 mA/W at 0.5 V), and 110 meV (LH-SO (split off), 3.7 mA/W at 4.3 V). In 2016, K. Gallacher et al. extended the mid-infrared inter-sub-band absorption with a p-Ge quantum well with Si_0.5_Ge_0.5_ to 6–9 μm [45].

The structure of SiGe-based intraband transition devices was modified in order to boost their performance. In 2016, Chong Wang et al. studied the structure of the Ge/Si_1−x_Ge_x_/Si dot-in-well (DWELL) (Figure 3e). The type-I heterojunctions would be formed with the SiO_2_ barrier layer and the Si well layer when the Si layer contacted the SiO_2_ layer, and they presented a structure of a Ge/Si system with a SiO_2_ barrier layer (Figure 3g). Due to the band offset, the DWELL structure was formed in the valence band. This structure could reduce the thermal excitation rate of the carriers and the dark current. It was more appropriate for constructing the resonant tunneling DWELL [46].

**Figure 3 materials-16-01562-f003:**
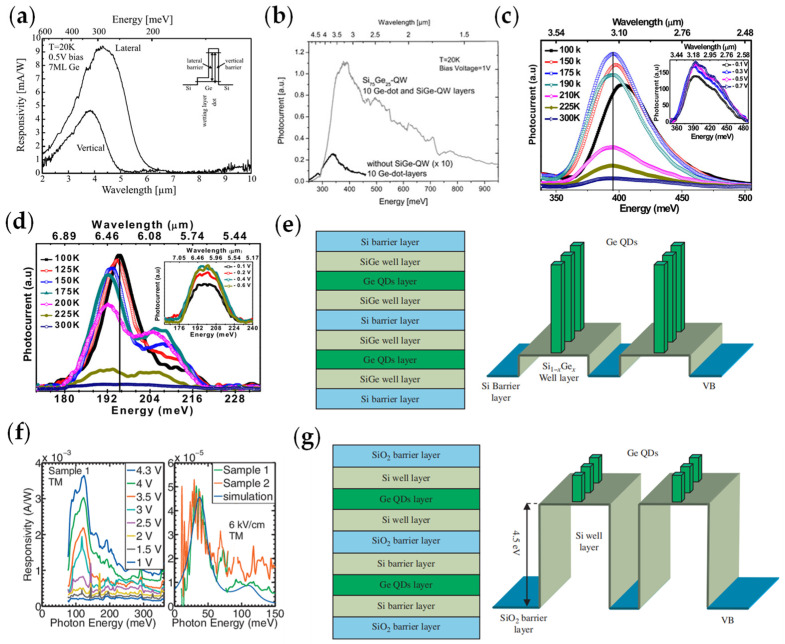
Structure and performance characterization of SiGe−based infrared intraband photodetectors: (**a**) Responsivity spectra of lateral and vertical infrared photodetectors with Ge dots. The right inset is a schematic of the excitation in the lateral and in the vertical cases [41]. Copyright 2001, Infrared Physics and Technology. (**b**) Photocurrent spectra of a Si/Ge/Si/SiGe multilayer structure and a sample containing no SiGe quantum wells [42]. Copyright 2003, Physical E. (**c**) Photocurrent spectra of capped Ge/Si QDs at the different temperatures in the near−infrared area. The inset shows photocurrent spectra under different bias conditions. (**d**) Photocurrent spectra of capped Ge/Si QDs at the different temperatures in the middle−infrared area. The inset shows photocurrent spectra under different bias conditions [43]. Copyright 2010, Applied Physics Letters. (**e**) The structure of the Ge/Si_1-x_Ge_x_/Si DWELL and the 3D energy band [46]. Copyright 2016, Journal of Nanoscience and Nanotechnology. (**f**) Responsivity peak for samples at high bias and transverse magnetic polarization based on LH−SO transition and terahertz responsivity peaks for samples with a low applied field of 6 kV/cm based on LH−LH transition [44]. Copyright 2011, Applied Physics Letters. (**g**) The structure of the Ge/Si/SiO_2_ DWELL and the 3D energy band [46]. Copyright 2016, Journal of Nanoscience and Nanotechnology.

Dark current and responsivity had always been important parameters of QWIP. To achieve a low dark current and high responsivity, on the one hand, the process of epitaxial growth was controlled to improve the uniformity of the structure [47,48,49]. On the other hand, Soumava Ghosh et al. improved the structure of SiGe/Si QWIPs and proposed the structure of GeSn/SiGeSn intraband QWIPs in 2021 [50]. The results showed that the optimized device structure achieved a low dark current of 2.35 pA at 2 V, the peak responsivity of 1.24 A/W at 4.3 µm, and the detectivity as high as 3.47 × 10^12^ Jones at 2 V and 77 K.

From the above analysis, it can be seen that the SiGe-based intraband transition photodetectors have a broader development prospect than the interband transition photodetectors. The SiGe-based intraband transition photodetector can be combined with other materials, which is expected to improve its responsivity and reduce the dark current.

### 2.2. ‘Bottom to Up’ Infrared Photodetectors

#### 2.2.1. HgSe CQD-Based Infrared Intraband Photodetector

QWIPs based on intraband transition are being developed and used in commercial detectors [10]. While the technology of QWIPs is still complicated, the development of new materials could provide a possible solution to solve this problem. For example, the chemical synthesis of CQDs is simple as the device substrates are flexible. The optical and electrical properties of CQDs can be tuned by controlling their physical dimensions. Therefore, in the following part, we mainly discuss the ‘bottom to up’ intraband photodetectors based on CQDs.

In 2014, Zhiyou Deng et al. [51] firstly developed the intraband photodetectors based on HgSe CQDs, which were illuminated with resonant 1S_e_-1P_e_ intraband absorption (Figure 4a). The intraband absorption peak was around 2000–3000 cm^−1^ on HgSe CQDs. The performance of the device is shown in Figure 4b–e. The dark current was minimal when the conduction band was filled with two electrons. The dark current could be decreased by a factor of 3200 with cooling from the temperature of 300 K to 80 K, indicating a better performance at low temperatures. The maximum responsivity was 1.2 × 10^−2^ A/W and the detectivity reached 8.5 × 10^8^ Jones at 80 K. Due to the narrow absorption and long intraband relaxation, it was determined that HgSe CQDs could be an ideal platform to test intraband photodetectivity.

Considering the large surface to volume ratio in nanomaterials, surface modification such as ligand modification would be vital. Adrien Robin et al. confirmed the effect of ligand modification on the performance of HgSe CQDs (Figure 5a) in 2016 [52], where they precisely controlled doping between 0.1 and 2 electrons per dot. Later in 2017, Bertille Martinez et al. measured the absolute energy levels of HgSe CQDs with different ligand exchanges (Figure 5b) [53]. In 2018, Bertille Martinez et al. proposed a method to graft functionalized polyoxometalate (POM) onto the HgSe CQDs’ surface (Figure 5c) [54]. To improve the affinity, the silyl-substituted POM was terminated with two thiol groups (TBA)_3_[PW_11_O_40_(SiC_3_H_6_SH_2_)] (POM-SH). More importantly, it confirmed that the dark current was decreased, and the activation energy was increased (Figure 5d), which could enhance the performance of intraband transition based on CQDs.

In 2020, Menglu Chen et al. proposed a hybrid ligand exchange to produce high-mobility HgSe CQDs films (HgSe/hybrid) and compared it with solid-state ligand exchange of ethanedithiol (HgSe/EDT) [55,56]. The mobility reached 1 cm^2^/Vs in 7.5 nm-diameter CQDs (Figure 5e,f), which was a 100-fold increase compared with HgSe/EDT films. Subsequently, Menglu Chen et al. investigated the effect of HgSe CQDs’ size distribution on mobility, the conductivity gap, and the intraband photoconduction [57]. The result showed that mobility was exponentially dependent on size dispersion and the intraband photoconductive properties could be enhanced by improving the size dispersion.

**Figure 5 materials-16-01562-f005:**
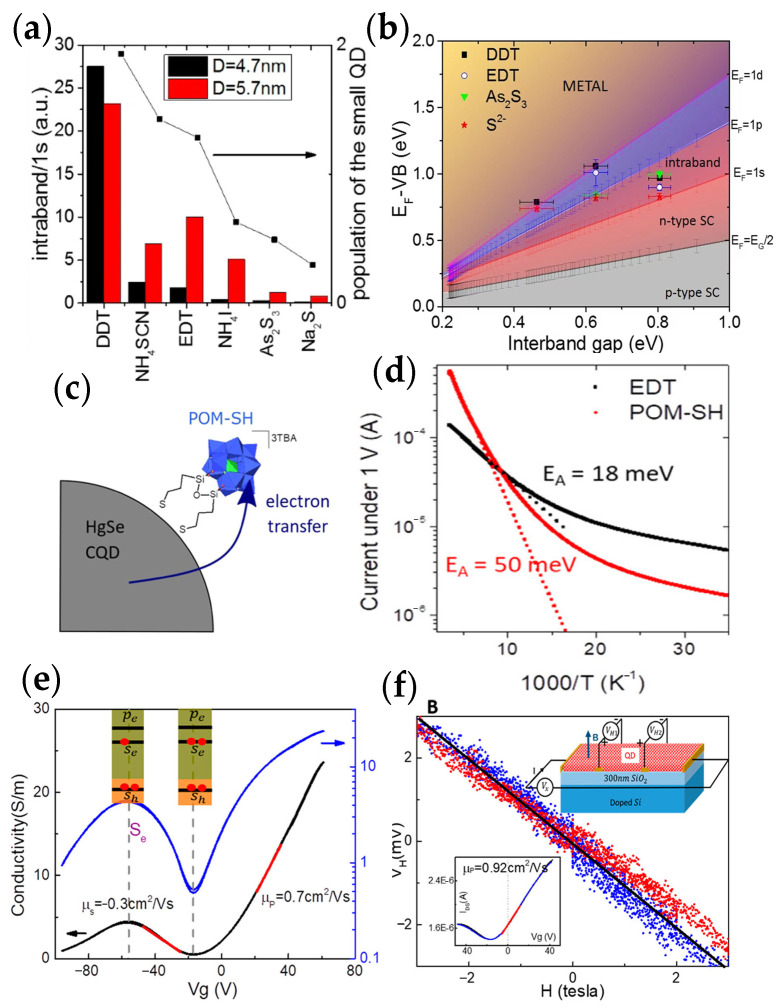
Performance of HgSe CQDs material: (**a**) Intraband signal of different ligands’ exchange [52]. Copyright 2016, ACS Applied Materials and Interfaces. (**b**) Fermi energy as a function of the interband gap [53]. Copyright 2017, ACS Applied Materials and Interfaces. (**c**) Schematic diagram of the HgSe CQDs with POM−SH ligands. (**d**) Current as a function of temperature [54]. Copyright 2018, The Journal of Physical Chemistry. (**e**) Field−effect transistor current of HgSe/hybrid at 80 K. (**f**) Hall voltage of HgSe/hybrid film at 200 K. The right inset shows the structure of the Hall device with a field−effect transistor, and the left inset shows the field−effect transistor transport curve [56]. Copyright 2020, The Journal of Physical Chemistry Letters.

To further improve the performance of intraband detectors based on HgSe CQDs, the modifications of material [58] and device structures were developed. In 2017, Xin Tang et al. designed a gold nano-disk array based on HgSe CQDs films and enhanced the photo-response via plasmon resonance [59]. As shown in Figure 6a, detectors for four bands (4.2 μm, 6.4 μm, 7.2 μm, and 9.0 μm) were integrated with the plasma nano-disk array. The responsivities reached 145 mA/W, 92.3 mA/W, 88.6 mA/W, and 86 mA/W.

In 2019, Clement Livache et al. proposed a mixture of HgTe and HgSe CQD infrared photodetectors, and they integrated the material into the photodiode (Figure 6b) [60]. The detectivity can reach 1.5 × 10^9^ Jones at 80 K, and the detectivity of this photodiode was two-fold higher than HgSe nanocrystals operating at the same temperature and wavelength. In 2021, Ananth Kamath et al. studied the performance of HgSe/CdS core/shell nanocrystals, which could achieve intraband optical absorption at 5 μm [61]. The CdS shells increased the intraband emission intensity. In 2022, Adrien Khalili et al. unveiled the coupling between HgSe and HgTe materials by using the mid-infrared transient reflectivity measurement (Figure 6c) [62]. This structure of HgSe and HgTe preserved the intraband absorption of HgSe, reduced the dark current, and boosted the time response. This hybrid material was coupled to a guided-mode resonator, resulting in a 4-fold enhancement of the photocurrent signal from the intraband contribution. The detectivity reached 10^9^ Jones at 80 K and 5 μm and the responsivity reached 3 mA/W with a weak temperature dependence. In comparison to other intraband photodetectors, its response time was below 200 ns.

Recently, there has been progress in the practical application of intraband HgSe CQD detectors. Menglu Chen et al. developed a high-performance intraband thermal imaging camera as well as a CO_2_ gas sensor with the range from 0.25 to 2000 ppm and the sensitivity of 0.25 ppm [63]. They utilized a mixed-phase ligand-exchange method at room temperature, which achieved high mobility on HgSe CQDs with the reduced surface ligand length. The response speed of their proposed intraband transition detector with high mobility and controlled doping reached several μs. It was improved by a factor of 1000-fold compared with the reference HgSe intraband detectors. The responsivity was 77 mA/W with 55-fold improvement compared to low-mobility devices. The detectivity achieved more than 1.7 × 10^9^ Jones at 80 K, more than one order higher compared with low-mobility devices. Thermal imaging is demonstrated in Figure 6d by the HgSe CQDs intraband transition-based detector.

According to the analysis of the HgSe CQD-based detector, the number of doped electrons in the conduction band of HgSe CQDs is one of the important factors affecting the dark current. Therefore, precisely controlling the number of electrons doped in the conduction band is one of the methods to solve the large dark current. In the future research, mixing HgSe CQDs with other materials is an important direction to improve the performance of HgSe CQDs.

#### 2.2.2. HgS CQD-Based Infrared Intraband Photodetector

Considering the similar band structure of mercury chalcogenides, Kwang Seob Jeong et al. studied the intraband absorption of HgS CQDs [64]. They firstly confirmed that stable carriers were present in the quantum state of strongly confined CQDs in ambient conditions. They investigated the intraband photoluminescence of zinc blend β-HgS (Figure 6h,i). In 2016, Guohua Shen et al. researched intraband transitions of air-stable n-doped HgS CQDs and HgS/CdS CQDs [65]. The large-sized HgS CQD could become a surface plasmon (Figure 6j) with an electron density of 1.6 × 10^19^ cm^−3^. This core/shell structure improved the thermal stability of the HgS cores. The research on HgS CQD materials expands the application of mercury series materials in intraband transition.

**Figure 6 materials-16-01562-f006:**
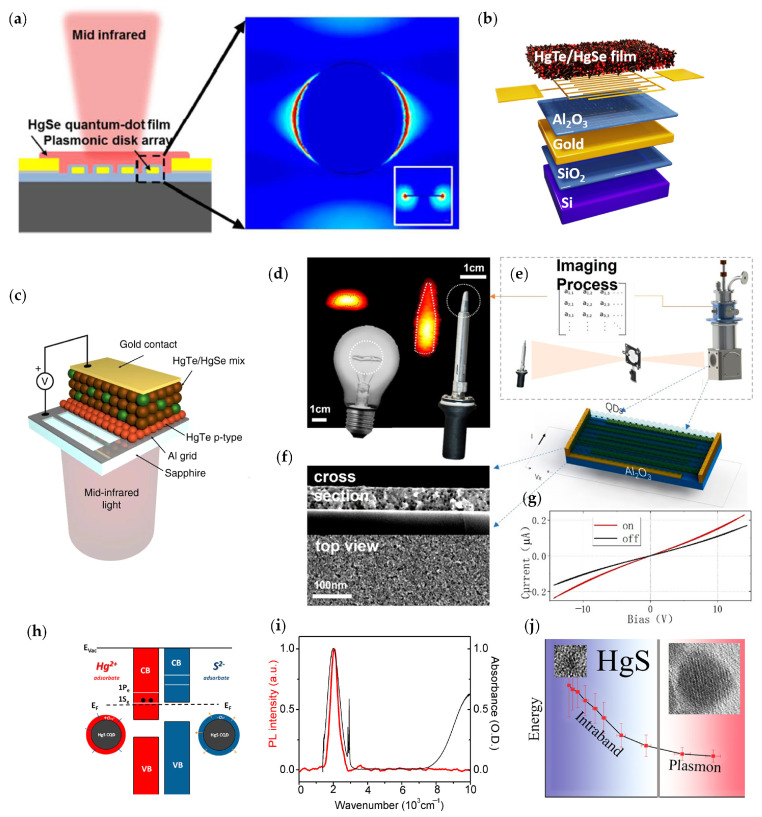
Structure and performance of HgSe and HgS CQD intraband photodetectors: (**a**) Schematic of HgSe CQD intraband photodetectors with plasmonic nano−disk array [59]. Copyright 2017, Journal of Materials Chemistry C. (**b**) Schematic diagram of the photodiode [60]. Copyright 2019, Nature Communications. (**c**) Schematic diagram of the guided−mode resonator device [62]. Copyright 2022, ACS Photonics. (**d**) Thermal imaging by high−mobility HgSe intraband photoconductor. (**e**) Schematic diagram of the imaging devices and structure diagram of the HgSe CQD photoconductor. (**f**) SEM of the HgSe CQD photodetector. (**g**) Current of the HgSe CQD photoconductor without (black line) and with (red line) blackbody radiation [63]. Copyright 2022, ACS Nano. (**h**) Schematic energy diagrams of the Hg^2+^− and the S^2−^−treated HgS CQDs. (**i**) Intraband photoluminescence emission (red line) and absorption spectrum (black line) [64]. Copyright 2014, The Journal of Physical Chemistry Letters. (**j**) Intraband absorption energy and plasmon energy of HgS CQDs as a function of particle sizes [65]. Copyright 2016, The Journal of Physical Chemistry C.

#### 2.2.3. Ag_2_Se CQD-Based Infrared Intraband Photodetector

The study of Ag_2_Se CQDs is attractive for the fabrication of non-toxic infrared photodetectors. In 2018, Mihyeon Park et al. proposed Ag_2_Se CQDs with steady-state intraband absorption and photoluminescence [66]. The spectroelectrochemistry (Figure 7c) and photocurrent were measured by using the Ag_2_Se nanocrystal/ZnO thin-film transistor (TFT) (Figure 7a). The Ag_2_Se nanocrystal/ZnO could be used in wavelength-selective infrared TFT photodetectors. At the same time, Junling Qu et al. investigated the photoconductivity of Ag_2_Se CQDs (Figure 7d) [67]. The responsivity had been estimated to be 8 μA/W.

To enhance the responsivity of Ag_2_Se CQDs, Shihab Bin Hafiz et al. firstly reported a vertical CQDs heterojunction device with Ag_2_Se/PbS/Ag_2_Se CQD stacks in 2021 (Figure 7e,f) [68], which reduced the dark conductivity and simplified the device fabrication procedures. The responsivity was increased by approximately 70 times compared with other lateral photoconductive photodetectors based on Ag_2_Se CQDs at 3–5 μm at room temperature, and the external quantum efficiency was 0.36% at 80 K. The device showed the detectivity of 3 × 10^5^ Jones at 300 K (Figure 7g).

In order to optimize the structure based on Ag_2_Se CQDs and PbS CQDs, in 2021, Shihab Bin Hafiz et al. reported p-n heterojunction diodes with strong rectifying characteristics based on Ag_2_Se CQDs and PbS CQDs (Figure 7h,i) [69]. The mixture of Ag_2_Se CQDs and PbS CQDs acted to block the transport of ground-state electrons and holes, as well as providing a flow of photoexcited electrons. The dark resistivity of this device was improved to 2 × 10^5^ Ω·cm, whereas the resistivity of the reference Ag_2_Se CQDs was only 1 × 10^3^ Ω·cm (Figure 7j). The peak responsivity was 19 mA/W at 4.5 μm, which was mainly contributed by the Ag_2_Se CQDs in the binary CQDs film. The detectivity was increased 30-fold compared with the reference Ag_2_Se CQDs, reaching 7.8 × 10^6^ Jones at 300 K. Experiments showed that the optimum mixing ratio was Ag_2_Se/PbS = 0.04.

The detectivity of Ag_2_Se CQD-based photodetectors can only reach 10^5^–10^6^ Jones, which is a big gap compared with other optoelectronic devices. Therefore, it is possible to consider mixing Ag_2_Se CQD-based photodetectors with other interband materials or adding optical structures to achieve improve the performance.

**Figure 7 materials-16-01562-f007:**
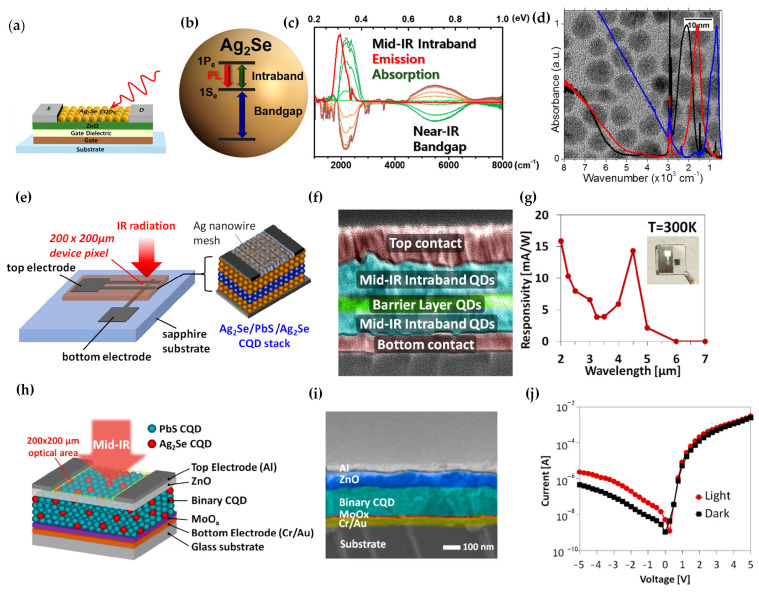
Structure and performance of Ag_2_Se CQD intraband photodetectors: (**a**) Schematic diagram of the Ag_2_Se CQD/ZnO field−effect transistor device. (**b**) Schematic of the Ag_2_Se CQDs band. (**c**) Differential absorption spectra and emission spectra of Ag_2_Se CQDs [66]. Copyright 2018, ACS Photonics. (**d**) Spectra of different−size Ag_2_Se CQDs [67]. Copyright 2018, The Journal of Physical Chemistry C. (**e**) Schematic of the device vertical structure of Ag_2_Se/PbS/Ag_2_Se CQDs. (**f**) The cross-sectional SEM of the device. (**g**) Responsivity as a function of wavelength. The illustration is a photograph of the device [68]. Copyright 2021, ACS Applied Materials and Interfaces. (**h**) Schematic of the PbS/Ag_2_Se CQDs device structure, which consisted of a glass substrate, bottom contact (Cr/Au), MoOx layer, PbS/Ag_2_Se CQD mixture layers, ZnO nanoparticle layer, and top contact (Al). (**i**) The cross−sectional SEM of the PbS/Ag_2_Se CQDs device. (**j**) Current characteristics under dark and infrared illumination [69]. Copyright 2021, ACS Applied Materials and Interfaces.

#### 2.2.4. Other Semiconductor CQD Infrared Intraband Photodetectors

There are also studies on infrared intraband photodetectors based on other semiconductor CQDs, such as CdSe, CdS, PbSe, PbS, and so on. Due to instability on pure doped CdSe CQDs, most studies focus on CdSe core/shell CQD devices. In 2008, Anshu Pandey et al. reported the slow intraband relaxation in CdSe/ZnSe core/shell CQDs [70]. The intraband relaxation significantly slowed down with the increase of ZnSe shell thickness. The absorption spectra of the different core/shell compositions are shown in Figure 8a.

In 2016, Kwang Seob Jeong et al. studied the performance of CdS CQDs, CdSe CQDs, and the core/shell structure [71], which showed an increase of the intraband photoluminescence with the increasing shell thickness (Figure 8b). In 2022, Rodriguez-Magdaleno et al. calculated the intraband absorption coefficient between the 1S and 1P of spherical CdSe/CdS/ZnSe CQDs [72]. As the inner shell size increased, the intraband coefficient redshifted by approximately 0.044 eV and 0.037 eV (Figure 8c).

Lead-based CQD infrared photodetectors are being investigated [73,74]. In 2018, Haipeng Lu et al. studied n-type doped PbSe CQDs with In^3+^ [75]. With increasing In^3+^ content, the interband first exciton transition gradually whitens, accompanied by size-dependent intraband absorption. In 2020, Inigo Ramiro et al. reported intraband absorption and photodetection capability of heavily doped PbS CQDs in the range of 5 to 9 μm (Figure 8d,e) [76]. The intraband response time was 30 ms, while the interband response time was 200 ms. The intraband responsivity was 1.5 × 10^−4^ A/W and the peak detectivity could achieve 4 × 10^4^ Jones at 80 K and 6.8 μm. Due to the large bandgap of the PbS bulk crystals, intraband transition-based detection reached spectral ranges that interband transition-based detection cannot. Detectors based on intraband transitions of PbS CQDs might be applied in CMOS-compatible low-cost multispectral imaging systems.

Lead-based CQDs have nearly 90% fluorescence quantum yield and high carrier mobility in the near-infrared region [77,78,79], but poor performance of intraband detection in the infrared region. For the intraband transition device based on lead-based CQDs, the use of core layer structures or ligand-exchange materials can be considered to improve the performance of the detectors.

#### 2.2.5. Perovskite-Based Infrared Intraband Photodetector

Perovskite-based intraband transition detectors are being investigated. Norah Alwadai et al. proposed a vertical photodetector (Figure 8f) based on Gd-doped ZnO nanorods/CH_3_NH_3_PbI_3_ perovskite on metal substrates in 2017 [80], which detected ultraviolet to infrared fields. The responsivity of white light reached 28 A/W and the responsivity of infrared illumination reached 0.22 A/W. The detectivities were 1.1 × 10^12^ Jones and 9.3 × 10^9^ Jones, respectively. The photodetector based on perovskite has been proven for the first time to extend its detection ability to the infrared band and has a high room temperature response in the detected spectrum.

The rapid relaxation of “hot” carriers (HCs) above the bandgap was the key factor limiting the efficiency of lead-halide perovskite (LHP) solar cells. In 2018, Thomas R. Hopper et al. reported the intraband cooling dynamics of five common LHPs (FAPbI_3_, FAPbBr_3_, MAPbI_3_, MAPbBr_3_, and CsPbBr_3_), and observed a cooling time of 100–900 fs [81] (Figure 8g,h). The cooling time began to slow with the increasing density of HCs. It was observed through experiments that the cooling dynamic of CsPbBr_3_ had the strongest dependence on the HCs’ density. They attributed this to the smaller specific heat capacity of all-inorganic perovskites due to the presence of fewer optical phonon modes. This work highlighted the critical role of cations in the HC kinetics of LHP and the importance of utilization of HCs in future photodetector applications.

In 2019, Benjamin T. Diroll studied the intraband relaxation of the MAPbI_3_ and MAPbBr_3_ polycrystalline thin films in the temperature range of 80 to 350 K [82] (Figure 8i). The intraband relaxation of such perovskite films was maintained at the sub-picosecond scale, which was comparable to typical semiconductors such as gallium arsenide and silicon. In the same year, Benjamin T. Diroll et al. studied the intraband relaxation of all-inorganic CsPbBr_3_ and hybrid organic–inorganic FAPbBr_3_ nanocrystals under the conditions of particle size, excitation energy, sample temperature, and excitation fluence [83]. In the single electron–hole pair area of each nanocrystal, the hot carriers in CsPbBr_3_ nanocrystals showed a slower cooling than FAPbBr_3_ nanocrystals. The experimental results demonstrated that the carrier cooling slowed down when each nanocrystal generated more than one electron–hole pair. The persistence of detectable hot carriers increased with the Auger recombination time in hybrid perovskite nanocrystals.

The perovskite-based intraband transition photodetectors have extended the range of photodetection to the ultraviolet-to-infrared region. In future research directions, perovskite-based intraband transition photodetectors should enhance their response speed in the infrared region [80].

**Figure 8 materials-16-01562-f008:**
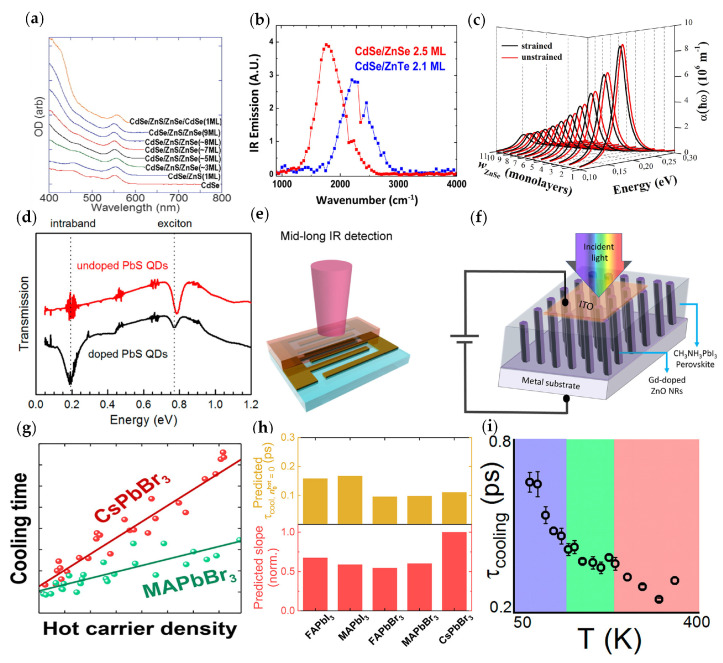
Structure and performance of CdSe, PbS CQDs, and perovskite intraband materials: (**a**) Absorption spectra at different core−shell compositions [70]. Copyright 2008, Science. (**b**) Photoluminescence of CdSe/ZnSe (red line) and CdSe/ZnTe (blue line) [71]. Copyright 2016, ACS Nano. (**c**) Intraband absorption coefficient of CdSe/CdS/ZnSe CQDs as a function of incident energy [72]. Copyright 2022, Materials Science in Semiconductor Processing. (**d**) Transmission spectra of undoped (red line) and doped (black line) PbS CQDs. (**e**) Schematic of a PbS CQDs device [76]. Copyright 2020, Nano Letters. (**f**) Schematic diagram of the photodetector based on perovskite material [80]. Copyright 2017, ACS Applied Materials and Interfaces. (**g**) Cooling time of CsPbBr_3_ and MAPbBr_3_ as a function of hot carrier density. (**h**) Comparison of the polaron scattering times and the inverse of the per−unit cell−specific heat capacity for the five perovskite material systems [81]. Copyright 2018, ACS Energy Lett. (**i**) Cooling time of MAPbI_3_ as a function of temperatures [82]. Copyright 2019, J Phys Chem Lett.

## 3. Conclusions and Perspectives

In this review, we have systemically summarized recent research progress on infrared photodetectors based on intraband transition. Precisely, we mainly classified the intraband infrared detectors, including ‘up to bottom’ devices such as QWIPs and ‘bottom to up’ devices such as QDIPs. Throughout the review, it can be seen that intraband photodetectors have gradually become the main trend in infrared detection. Nevertheless, photodetectors based on intraband transitions still have problems, such as a large dark current and low responsivity, compared with those based on interband transitions. There are some future challenges, as described below:(1)The working temperature of detectors based on intraband transition needs be improved. Most of the existing photodetectors for intraband transition work in integrated Dewar structures, which allow infrared photodetectors to be isolated from the outside air and operate at low temperatures. Bare infrared photodetectors are easily disturbed by temperature and oxygen in the air. Among them, the analysis of QDIPs and QWIPs shows that most photodetectors with intraband transitions need to be lowered to lower temperatures for high performance. Therefore, research on high operating temperature infrared intraband photodetectors would be one of the main fields.(2)Non-toxic infrared photodetectors would be another hot field. Most intraband transition materials are still based on heavy metals, such as mercury and lead. The toxicity of infrared materials makes infrared photodetectors rarely used in biomedical fields. The development of non-toxic materials such as Ag_2_Se CQDs would fill this part of the gap. However, the Ag_2_Se CQDs currently have low responsivity. It should be considered to chemically modify the surface of Ag_2_Se CQDs or combine it with other materials to improve the responsivity of Ag_2_Se CQDs. Therefore, it is imperative to explore more non-toxic intraband transition-based infrared photodetectors and improve their performance.(3)The fabrication process of detectors based on intraband transition needs to be improved, including the method of material growth, device uniformity, and repeatability. The uniformity of the device can affect the responsivity and detectivity of the detector. Among them, ‘up to bottom’ infrared intraband photodetectors have poor reproducibility, and most of them can only be tested in the laboratory environment. Efforts are needed to further improve the fabrication process of intraband infrared photodetectors to enable their commercial application. It should be considered to investigate automated or semi-automated equipment to fabricate photodetector films to reduce the effects of human interference.(4)The detectivity of infrared photodetectors should be improved and the dark current should be further reduced. Compared with the infrared detector of the interband transition, the intraband infrared detector has the disadvantages of low specific detectivity and a large dark current. On the one hand, the quantum efficiency of photodetectors is improved by modifying device structures, such as adding optical resonators and optical lenses. On the other hand, the infrared-sensitive materials need to be modified. For example, suitable ligand materials are selected for ligand exchange on CQDs. In the synthesis of CQDs, most of the solvents are oleylamine or octadecene, which are long-chain ligands. However, long-chain ligands will increase the signal-to-noise ratio of the device. Moreover, the mobility of the material of the carriers can be increased by ligand exchange. In particular, mixing intraband transition-based CQDs with interband transition-based CQDs or forming a core/shell structure can improve the performance of intraband QDIPs.(5)Development of infrared detectors for multi-color intraband transition. At present, the research on multi-color infrared photodetection is increasing. The multi-color photodetector can obtain spectral information of different bands at the same time, which can significantly improve the ability of target identification and imaging. However, the research on multi-color intraband infrared photodetectors is still limited. According to the analysis of intraband transition-based photodetectors, it can be seen that most of the photodetectors use single-band detection. The multi-color detection function can be realized by integrating intraband transition materials of different wavelengths. At the same time, it is necessary to extend the detection range of the intraband infrared photodetectors. At present, intraband transition-based photodetectors are rarely used in the long-wave range, and the next step would be to develop long-wave detection bands.(6)The development of large focal plane array techniques based on intraband transition needs be enhanced. Most of the existing intraband infrared photodetectors are still single-pixel photodetectors, which would limit their applications. The focal plane array infrared intraband photodetectors are the main direction of future developments, and the readout circuit of the focal plane array should be further improved to achieve the purpose of real-time and high-speed readout signals.

## Figures and Tables

**Figure 1 materials-16-01562-f001:**
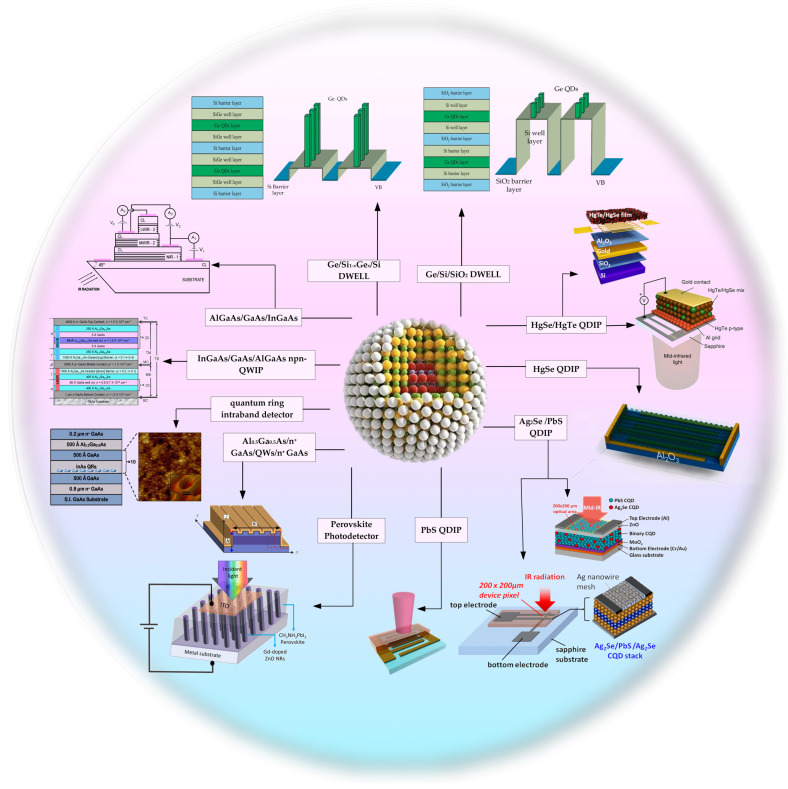
Structure diagram of the infrared photodetectors based on the intraband transition.

**Figure 4 materials-16-01562-f004:**
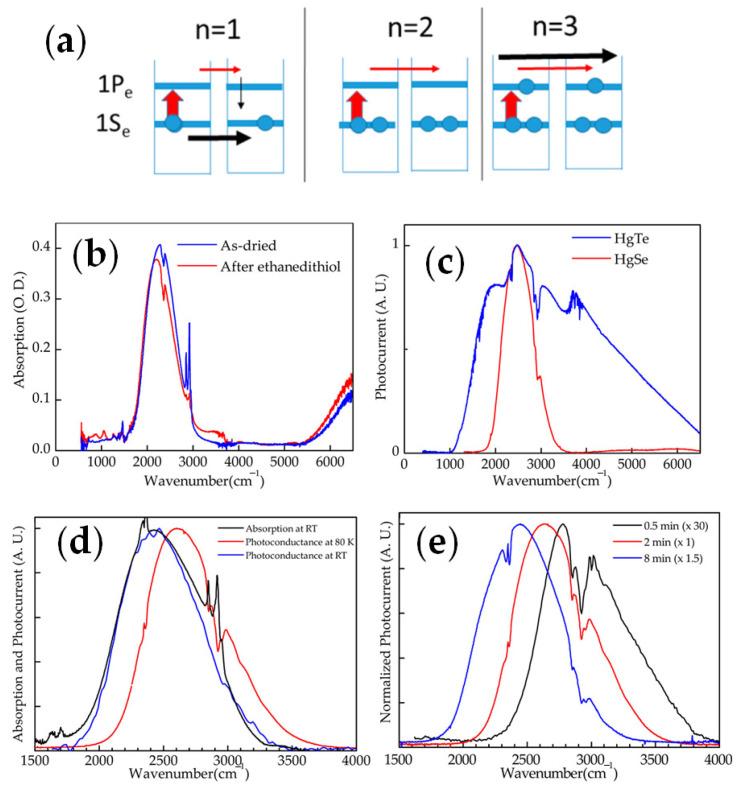
Characterization of HgSe CQDs material: (**a**) Schematic of doping electrons in intraband conduction. (**b**) Absorption spectra of HgSe CQDs on the ZnSe ATR window, deposited and dried from hexane:octane solution and exchanged by ethanedithiol. (**c**) Photocurrent spectra of the HgSe film and the HgTe film at 80 K. (**d**) Normalized absorption and photocurrent of HgSe film at room temperature and the photocurrent of HgSe film at 80 K. (**e**) Normalized photocurrent of HgSe film at various reaction times at 80 K with a 10 V bias [51]. Copyright 2014, ACS Nano.

**Table 1 materials-16-01562-t001:** Progress in the development of InAs-based devices with intraband transition.

Year	Materials	Detection Range (μm)	Detectivity (Jones)	Responsivity	Ref
2001	InAlAs/InP	10–18	--	--	[31]
2001	InAs/GaAs	0.5	10^9^	1 A/W	[32]
2002	InAs/AlGaAs	6.2	10^10^	14 mA/W	[33]
2004	InAs/InGaAs/GaAs	9.3	3 × 10^11^	0.71 A/W	[34]
2005	GaAs	0.8 57	6 × 10^9^ 5 × 10^9^	7 A/W	[35]
2007	InGaAs/InAlAs GaAs/AlGaAs	4.3 9.6	-- --	-- --	[36]
2013	InP/In_0.48_Ga_0.52_As	9	5 × 10^10^	1.5 A/W	[27]
2015	AlGaAs/GaAs	4.2	--	1.7 × 10^4^ V/W	[37]
2016	InGaAs/InAlAs	2.11	--	--	[38]
2018	In_0.5_Ga_0.5_Sb/InAs	3.81	2.4 × 10^9^	0.71 A/W	[39]
2020	Al_0.3_Ga_0.7_As/GaAs	6.6	1.8 × 10^10^	0.8 A/W	[40]

## Data Availability

Not applicable.

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
