# Peer review of "The Historical Development of Infrared Photodetection Based on Intraband Transitions"

_materials, 2023, doi:10.3390/ma16041562_

Round 1
Reviewer 1 Report
In this review paper, the authors discuss the recent developments on intraband infrared photodetectors, including ‘bottom to up’ devices like quantum well devices based on the molecular beam epitaxial approach, as well as ‘up to bottom’ devices like colloidal quantum dot devices based on the chemical synthesis. The topic itself is very interesting, and meets the scope of the Materials journal. The manuscript is well organized with solid data analysis and clearly presented. Conclusions are convincing. Recommend to accept for publication, after some minor revision as follows:
The authors refer to the works on Ge/Si based infrared intraband photodetectors in a quite limited view. I recommend to cover this topic more broadly and deeply.
Reviewer 2 Report
This is quite good overview for researchers who are starting research in the field of photodetection based on intraband forefronts in semiconductors.
The introduction and conclusion are well written in the paper, but the deductions formulated in the conclusion are practically not connected with the main text of the article. The style of the material in the main text is a chronological (historical) description of the most important experimental results available in modern literature. This is also confirmed by the analysis of cited literature: half of the references are 10 or more years old, the other half are publications of the last five years. At the same time, the authors practically do not provide an analysis of the results used. In this regard, it would be logical to change the title of the article from "Infrared photodetection based on intraband transition", for example, to "The historical development of infrared photodetection based on intraband transitions".
In addition, there are a number of questions and comments.
1. The terms "up to bottom" and "bottom to up" are used in different spheres of activity and have different meanings. Authors should explain what content they put into these terms in this article.
2. Figures 1 and 2 are vastly overladen - it's almost impossible to read some of the inscriptions on these pictures. Attempts to enlarge the images do not help - there is not enough resolution.
3. On page 6 (lines 193-195) in the sentence "In 2014, Zhiyou Deng et al. [41] firstly developed the intraband photodetectors based on HgSe CQDs, which were illuminated with resonantly with 1Se-1Pe intraband absorption (Figure 4a)”, is an unnecessary preposition in my opinion.
4. In the caption to Fig. 4 (page 7, line 209): “…at various reaction times…”, it is not clear which times are meant.
5. On page 12, line 370: typo - instead of "lithium halide perovskite (LHP)" should be "lead-halide perovskite".
6. On page 14, line 438: typo - instead of "olamine" should be "oleylamine".
7. On page 15, line 509: in reference 23: Perera, A. G. U.; Aytac, Y.; Ariyawansa, G.; Matsik, S. G.; Buchanan, M.; Wasilewski, Z. R.; Bhowmich, S.; Huang, G.; Guo, W.; Lee, C. S.; Bhattacharya, P.; Liu, H. C. In Photo detectors for multi-spectral sensing, 2011 11th IEEE International Conference on 508 Nanotechnology, 15-18 Aug. 2011; 2011; pp 286-291.- there are unnecessary dates.
More precisely, the link looks like this:
A. G. U. Perera, Y. Aytac, G. Ariyawansa, S. G. Matsik, M. Buchanan, Z. R. Wasilewski, S. Bhowmich, G. Huang, W. Guo, C. S. Lee, P. Bhattacharya, H. C. Liu. Photo Detectors for Multi-Spectral Sensing, 2011 11th IEEE International Conference on Nanotechnology, Portland Marriott,
August 15-18, 2011, Portland, Oregon, USA, p. 286-291.
8. On page 16, lines 568-570: in reference 51 there are incomplete data: Khalili, A.; Weis, M.; Mizrahi, S. G.; Chu, A.; Dang, T. H.; Abadie, C.; Grйboval, C.; Dabard, C.; Prado, Y.; Xu, X. Z.; Pйronne, E.;
Livache, C.; Ithurria, S.; Patriarche, G.; Ramade, J.; Vincent, G.; Boschetto, D.; Lhuillier, E., Guided-Mode Resonator Coupled with Nanocrystal Intraband Absorption. ACS Photonics 2022.
Full link is: Khalili, A.; Weis, M.; Mizrahi, S. G.; Chu, A.; Dang, T. H.; Abadie, C.; Grйboval, C.; Dabard, C.; Prado, Y.; Xu, X. Z.; Pйronne, E.; Livache, C.; Ithurria, S.; Patriarche, G.; Ramade, J.; Vincent, G.; Boschetto, D.; Lhuillier, E., Guided-Mode Resonator Coupled with Nanocrystal Intraband Absorption. ACS Photonics 2022, 9, (3), 10.1021/acsphotonics.1c01847.
9. In my opinion, the authors did not complete the literature search. My brief search had found three works not mentioned by the authors:
- Thibault Apretna, Sylvain Massabeau, Charlie Gréboval, Nicolas Goubet, Jérôme Tignon, Sukhdeep Dhillon, Francesca Carosella, Robson Ferreira, Emmanuel Lhuillier and Juliette Mangeney*. Few picosecond dynamics of intraband transitions in THz HgTe nanocrystals // Nanophotonics 2021; 10(10): 2753. https://doi.org/10.1515/nanoph-2021-0249
- Xue Zhao, Ge Mu, Xin Tang ,Menglu Chen. Mid-IR Intraband Photodetectors with Colloidal Quantum Dots // Coatings 2022, 12(4), 467.
https://doi.org/10.3390/coatings12040467
J.-Z. Zhang and I. Galbraith. Intraband absorption for InAs/GaAs
quantum dot infrared photodetectors //Appl. Phys. Lett. 84, 1934, (2004);
https://doi.org/10.1063/1.1687459.
I think that a thorough search will allow them to find more additional publications on the topic of the article.
10. The authors should carefully reread their text and make appropriate corrections.

Reviewer 3 Report
Accept in present form.
Reviewer 4 Report
This review addresses the basics, science, engineering and technology of IR photodetectors based on intraband transition. The idea of this review is appealing, and its scope is reasonable. However, we believe that a good review article, in addition to a review/summary of the published results, shall offer also critical/unique perspectives, comments, and suggestions from the authors on the critical issues relevant to the topic. The authors offered some comments addressing the above suggestions and provide a few remarks in the Conclusion and Perspective section, however, is limited by what is already in the literature. Stronger emphasis and discussion of selected unique scientific perspectives, the future of R&D directions and other relevant perspectives on the topic based on the author’s research perspective of experts in the field will strengthen the review impact considerably.
Suggested articles to include in this review is shown below. The authors shall clearly differentiate this review from one already published.
Interband Quantum Cascade Infrared Photodetectors: Current Status and Future Trends, P. Martyniuk, A. Rogalski, and S. Krishna, Phys. Rev. Applied 17, 027001 – Published 11 February 2022
In summary, minor revisions are recommended before further considerations will be offered.
Round 2
Reviewer 1 Report
The manuscript has been sufficiently improved to warrant publication in Materials.Reviewer 2 Report
All my comments have been taken into account by the authors and appropriate corrections have been made. There are no new comments.
Reviewer 4 Report
The authors have addressed my concerns and revised the manuscript accordingly or provided reasonable rebuttal. Thus, I recommend this review be published as is. Thank you.